

# 25-gauge vitrectomy with gas tamponade for uncomplicated rhegmatogenous retinal detachment: experienced *vs.* inexperienced surgeons

Martin Pencak[1], Zbynek Stranak[1], Jakub Dite[1], Jana Vranova[2], Pavel Studeny[1] and Miroslav Veith[1]

[1] Department of Ophthalmology, Third Faculty of Medicine, Charles University, and University Hospital Kralovske Vinohrady, Prague, Czech Republic

[2] Department of Medical Biophysics and Medical Informatics, Third Faculty of Medicine, Charles University, Prague, Czech Republic

## ABSTRACT

**Background**. Rhegmatogenous retinal detachment (RRD) is a vision-threatening condition that requires prompt surgical intervention. With advancements in surgical techniques and equipment, pars plana vitrectomy (PPV) has become increasingly popular for the management of RRD. This, in turn, requires beginner vitreoretinal surgeons to be able to manage RRD early in their training.

**Introduction**. Comparison of the results and complication rates of a 25-gauge (25 g) PPV with gas tamponade for RRD between experienced and inexperienced surgeons.

**Methods**. This is a retrospective comparative consecutive case series study of patients with uncomplicated RRD treated with 25 g PPV with gas tamponade. Patients were divided into two groups: in the experienced surgeon group (ESG), the procedure was performed by an experienced vitreoretinal surgeon, in the inexperienced surgeon group (ISG), it was performed by two inexperienced surgeons. Anatomical and functional results and complication rates were compared between the two groups.

**Results**. A total of 216 eyes were included in the study. In the ESG (106 eyes), the single operation success rate was 94.3%, and the final success rate was 100%. In the ISG (110 eyes), the single operation success rate was 93.6%, and the final success rate was 100%. The difference in single surgery success rate between groups was not statistically significant ($P = 0.828$). The mean postoperative BCVA improvement was 0.348 decimal in the ESG and 0.405 decimal in the ISG ($P = 0.234$). The difference in complication rates between groups was not significant.

**Conclusions**. A 25 g PPV with gas tamponade for the treatment of RRD yields excellent anatomical results and improvement in best-corrected visual acuity (BCVA). With good technique and the use of modern vitrectomy machines and instruments, some inexperienced surgeons can achieve a high single-surgery success rate, suggesting a short learning curve. The complication rate is comparable between experienced and inexperienced surgeons.

Corresponding author
Martin Pencak, pencak@volny.cz

## INTRODUCTION

Rhegmatogenous retinal detachment (RRD) is a separation of the neurosensory retina from the retinal pigment epithelium caused by the influx of subretinal fluid through the retinal break (RB). Vitreous traction plays an important role in the creation of RB and the formation of RRD (*Mitry et al., 2010*). It is a vision-threatening condition that requires prompt surgical intervention. Several surgical techniques for the treatment of RRD have been developed, and scleral buckling (SB), pneumatic retinopexy (PnR), and pars plana vitrectomy (PPV) are currently used (*Escoffery et al., 1985*; *Hilton & Grizzard, 1986*; *Chrapek et al., 2013a*; *Chrapek et al., 2013b*; *Hejsek et al., 2014*; *Veith et al., 2018*).

PnR is a procedure that involves applying cryotherapy or laser retinopexy around the RB before or after the intravitreal injection of expansile gas. It is a less invasive procedure than PPV or SB; however, it is suitable only for uncomplicated retinal detachments and requires careful patient selection to achieve good results (*Hillier et al., 2019*).

In SB, silicone exoplants are used to indent the sclera over the RB previously treated with cryopexy or laser retinopexy, thus lessening the vitreous traction. It is an extraocular procedure with advantages over PPV, especially in young, phakic patients with clear optical media and without posterior vitreous detachment (PVD). It can be combined with subretinal fluid drainage and intravitreal injection of expansile gas (*Park, Lee & Lee, 2018*).

PPV is an intraocular surgery aimed at removing the vitreous, draining the subretinal fluid and treating the RB with cryopexy or laser retinopexy. Air, mixtures of expansile gases or silicone oil are used as a tamponade to block the RB for the time of chorioretinal scar formation (*Escoffery et al., 1985*). PPV can be combined with SB, although whether this affects the outcome remains uncertain (*Nichani et al., 2022*; *Rajsirisongsri et al., 2024*). The development of sutureless, small-gauge vitrectomy and advancements in surgical techniques and equipment have made PPV increasingly popular among surgeons in the management of RRD (*McLeod, 2004*; *Ho et al., 2008*). Thus, it is becoming increasingly important even for beginner vitreoretinal surgeons to manage retinal detachment cases successfully.

Several studies have compared the results of PPV for retinal detachment between experienced and inexperienced surgeons (*Heimann et al., 2006*; *Dugas et al., 2009*; *Ehrlich et al., 2011*; *Mazinani et al., 2012*; *Keller, Haynes & Sparrow, 2016*; *Sallam et al., 2018*). In the abovementioned studies, the retinal reattachment success rates were comparable between experienced and inexperienced vitreoretinal surgeons. However, all the abovementioned studies have some limitations. Some of them use older generations of vitrectomy machines or contact lenses for visualisation, and their results do not reflect the advancements in surgical instruments and machines (*Heimann et al., 2006*; *Dugas et al., 2009*; *Ehrlich et al., 2011*; *Mazinani et al., 2012*). Some focus on operation success rates and visual acuity change, but do not report on complication rates (*Dugas et al., 2009*; *Mazinani et al., 2012*; *Keller, Haynes & Sparrow, 2016*; *Sallam et al., 2018*). The study by *Sallam et al. (2018)* is a national clinical database study and does not include details about the surgical techniques or instruments used for the PPV. The study by *Heimann et al. (2006)* focuses primarily on

the risk factors for primary surgery failure, with surgeon experience being just one of the studied variables.

In our study, we compared the anatomical success rate, change in visual acuity, employed surgical techniques, and complication rates in patients with RRD, performed using a 25-gauge (25 g) PPV with gas tamponade without scleral buckling, between an experienced surgeon and two inexperienced surgeons using a high-speed vitrectomy probe and noncontact wide-angle visualisation system. This article was published as a preprint (*Pencak et al., 2020*).

## MATERIALS AND METHODS

### Patient selection

Ours was a single-centre retrospective comparative consecutive case series study. Patients with RRD treated with 25 g PPV with gas tamponade without scleral buckling between October 2015 and June 2018 in the Department of Ophthalmology of the University Hospital Kralovske Vinohrady, Prague, were included in the study. Both pseudophakic and phakic patients were included. We excluded patients with proliferative vitreoretinopathy (PVR) grade C and higher, patients with a previous perforating eye injury and patients with a follow-up period shorter than 3 months. Patients were then divided into two groups. The first group (the experienced surgeon group (ESG)) included patients treated by an experienced surgeon (MV, 1,909 PPV at the beginning and 2,438 PPV at the end of the inclusion period). The second group (the inexperienced surgeon group (ISG)) included patients treated by two inexperienced surgeons (MP and ZS, 28 and 22 PPV, respectively, at the beginning and 172 and 161 PPV, respectively, at the end of the inclusion period). Both inexperienced surgeons started the training at the same time, were trained by the same surgeon (MV) and assisted and practised different steps in PPV surgeries for 2 years under the close supervision of the training surgeon before their first complete vitrectomy. This meant spending one day a week in the operating theatre assisting the experienced surgeon in approximately five PPV. Both had no previous experience with intraocular surgery, including cataract surgery. At that time, MV was the only experienced vitreoretinal surgeon at the University Hospital Kralovske Vinohrady Department of Ophthalmology. MP and ZS were the only two doctors undergoing training in vitreoretinal surgery. We compared the anatomical success rate, change in visual acuity, employed surgical techniques and complication rates between groups. All patients signed an informed consent before the surgery. The ethics committee waived the need for specific informed consent with participation in the study due to the retrospective nature of the study. The study protocol adhered to the tenets of the Declaration of Helsinki Principles. Approval was granted by the Ethics Committee of the Third Faculty of Medicine, Charles University, Prague and the Kralovske Vinohrady University Hospital, Prague, under the number EK-R/01/0/2020.

### Pre- and postoperative assessment

In all patients, the pre- and postoperative best-corrected visual acuity (BCVA) was assessed using Early Treatment Diabetic Retinopathy Study (ETDRS) charts, which were then converted to decimal values for statistical analysis. BCVA of counting fingers, hand motion,

or light perception were converted to decimal values using the chart published by *Holladay (2004)*. Pre- and postoperative slit-lamp examination and fundus biomicroscopy were performed to determine the extent of RRD, the location and number of RB, grading of the preoperative PVR, and to assess the postoperative state of the retina. Intraocular pressure (IOP) was measured using non-contact tonometry using an NT-530 (Nidek, Aichi, Japan) preoperatively, on the first postoperative day, and at all visits during follow-up. Hypotony was defined as an IOP < 10 mmHg, and hypertension as an IOP > 25 mmHg. Medical history was collected with emphasis on duration of symptoms of retinal detachment (floaters, flashing lights, curtain covering part of the visual field, visual acuity decline). The primary endpoint was the single surgery retinal reattachment rate. The secondary endpoints were the postoperative change in BCVA (in decimal) and complication rates. All postoperative endpoints were assessed on the last day of follow-up.

## Surgical technique

All patients underwent a three-port 25 g PPV, using the oblique cannula insertion technique described previously (*Hsu et al., 2008*), using a Constellation® vitrectomy machine (Alcon, Fort Worth, TX, USA), with an Ultravit® vitrectomy probe with a cutting rate of 5,000 cuts/min. Valved cannulas were used in all patients starting in August 2016. A Resight® 500 (Zeiss, Oberkochen, Germany) wide-angle visualisation system was used to visualise the fundus. Cannulas were placed 3.5 mm posterior to the limbus in pseudophakic eyes and 4.0 mm posterior to the limbus in phakic eyes. IOP was set to 25 mmHg during the surgery for both the fluid and air. Alternative IOP setting was set to 50 mmHg for fluid and 40 mmHg for air, however, increased fluid IOP was used only rarely and for short periods of time, for example, to stop retinal bleeding or in case of hypotony during surgery with normal IOP setting. An alternative IOP setting for air was used during fluid-air exchange to prevent ocular hypotony. A core vitrectomy was performed, followed by peripheral vitreous removal assisted by scleral indentation with the light probe. The light probe was used to indent the sclera, and peripheral vitreous removal was done in retroillumination. Shaving of the vitreous base was performed around RB and suspicious lesions. Anterior hyaloid dissection was not performed in phakic patients. Betamethasone suspension was used to stain the vitreous in some cases to allow for better visualisation of the vitreous, especially in cases where posterior vitreous attachment was suspected. Perfluorodecalin (Arcaline, Arcadophta, France) was used in some patients to immobilise a detached retina and facilitate peripheral vitreous removal. If an epiretinal membrane (ERM) or macular hole (MH) was present in the macula, brilliant blue (Ocublue, Aurolab, Madurai, India) dye was used, and ERM and internal limiting membrane (ILM) peeling was performed. Perfluorodecalin was used in patients with a detached macula when membrane peeling was required. Fluid-air exchange with subretinal fluid (SRF) drainage through a peripheral RB was performed using a Charles Flute Cannula (Alcon, Fort Worth, TX, USA) or a vitrectomy probe. Perfluorodecalin was used, or posterior retinotomies were performed, to achieve complete SRF drainage in patients where the surgeon was concerned about the risk of a retinal fold forming in the macula. Posterior retinotomies were also performed in patients where pre-existing RBs were not identified during vitrectomy. Perfluorodecalin

was used, and retinotomies were performed at the discretion of the surgeon. Complete SRF drainage was not required. Retinopexy of the margins of RB, lattice degenerations and other suspicious peripheral lesions were performed under air using an endolaser or cryotherapy probe. In some patients, retinopexy to the extent of the detached retina or a 360° retinopexy was performed. The extent of retinopexy depended on the number and location of RBs and the lattice degenerations, and was also at the surgeon's discretion. A non-expansive 20% mixture of sulfur hexafluoride (SF6) (Alchimia, Padua, Italy) or a 15% mixture of perfluoropropane (C3F8) (Alchimia, Padua, Italy) was used as a tamponade in all patients. The decision on which gas to use depended mainly on the location of RB. In patients with superior RB, 20% SF6 was generally used; in patients with inferior RB, 15% C3F8 was generally used. However, other factors were considered, like the presence and location of lattice degenerations and other suspicious peripheral lesions, patients' ability to posture after the operation, *etc.* The final decision on the gas tamponade was made solely by the surgeon. After the air-gas exchange, cannulas were removed, and the tightness of the sclerotomies was checked. If leakage was present, a digital massage of the sclerotomy was performed. If leakage persisted after digital massage, the sclerotomy was sutured using Vicryl 8-0 (Ethicon, Johnson & Johnson Int., New Brunswick, NJ, USA). After all the cannulas were removed, digital palpation was used to check the IOP at the end of the operation. If the IOP was considered low, additional gas mixture was injected through the sclera using a 30-gauge needle. No scleral buckling was performed. Depending on the location of the RB, patients were instructed on proper head positioning for one week after the operation. All patients were also advised on the normal postoperative course of events and were encouraged to come for a check-up immediately in case of abnormal development.

## Statistical analysis

Quantitative variables—age, follow-up period, pre- and postoperative BCVA, BCVA change, number of RB, extent of retinal detachment, duration of symptoms and time to surgery were tested using the Kolmogorov–Smirnov test on normality. Age and BCVA change were found to have normal distribution. They are given as means and standard deviations. To compare the age of patients and BCVA change after operation between both groups, the independent Student's $t$-test was calculated. Follow-up period, pre- and postoperative BCVA, number of RB, extent of retinal detachment, duration of symptoms and time to surgery were found to have non-normal distribution and are given as median and interquartile range (IQR). The Mann–Whitney $U$ test was used to compare these variables between the two groups. To compare the BCVA before and after surgery, the paired samples Wilcoxon test was used for all three surgeons together and for each surgeon separately. Retinal attachment success rates, lens status, state of the macula, complication rates, surgical techniques employed, and the number of patients with inferior RB in the ESG and ISG are given as total frequency and percentages. The differences in retinal attachment success rates, lens status, state of the macula, complication rates, employed surgical techniques and the number of patients with an inferior RB between the ESG and ISG were analysed using contingency tables—the Pearson's chi-square test or in the case of

small sample numbers, the exact Fisher test was calculated. Statsoft STATISTICA version 9 was used for statistical analysis. *P*-values less than 0.05 were considered to be statistically significant.

## RESULTS

We examined the documentation of 407 PPV for retinal detachment performed on 368 eyes between October 2015 and June 2018 in the Department of Ophthalmology of the University Hospital Kralovske Vinohrady, Prague. 245 PPV were performed by the experienced surgeon, 162 by the inexperienced surgeons. The reason for exclusion from the study was the use of silicone oil tamponade in 53 PPV performed by the experienced surgeon and in 21 PPV performed by the inexperienced surgeons, other inclusion and exclusion criteria were not met in other cases. In the end, 216 eyes of 216 patients met all the inclusion and exclusion criteria and were included in the study, 106 in the ESG and 110 in the ISG. Table 1 shows the baseline demographic and clinical characteristics of the participants.

The date of diagnosis of retinal detachment was recorded in 48 patients in both groups. The median time to surgery was 2.0 (IQR 0–3) days in ESG and 1.0 (IQR 0–3) day in ISG ($P = 0.435$). The duration of symptoms was recorded in 81 patients in ESG and 79 patients in ISG. The median duration of symptoms was 5.0 (IQR 0–10) days in both groups ($P = 0.889$). One patient in the ESG underwent unsuccessful pneumatic retinopexy with laser barricade around the RB before the PPV, and one patient had undergone scleral buckling for retinal detachment 11 years before the PPV. In the ISG, two patients underwent unsuccessful laser retinopexy around the RB before the PPV. There were two patients with high myopia in both ESG and ISG. There was no patient with a giant retinal tear in either group. A macular hole was present preoperatively in two patients in the ESG. ILM peeling was performed during PPV for retinal detachment in both patients, and the macular hole was closed in one of them. Table 2 shows the surgical techniques and gas tamponade used in both groups.

The results are shown in Table 3. The difference in single surgery success rates between individual surgeons was not statistically significant. After the second surgery, the success rate was 100% in the ESG and 97.3% in the ISG; a 100% success rate was achieved in the ISG after the third surgery. The causes of failure were RB in the scar after cryopexy in four patients in the ESG and two patients in the ISG, reopening of the original RB in one patient in the ISG, a newly diagnosed RB in three patients in the ISG, and a PVR in two patients in the ESG and in one patient in the ISG.

Of the 13 patients who required a second surgery for retinal redetachment, gas tamponade was used in 10, and silicone oil was used in three. PVR dissection was performed in two patients and retinectomy in one patient during the second surgery. Silicone oil was used as a tamponade for all three patients who required a third surgery. PVR dissection was performed in one patient and retinectomy in two patients during the third surgery. In one patient who required three surgeries, silicone oil was left in the eye as a permanent tamponade due to hypotony. Scleral buckle was not used in any of the patients. The

**Table 1  Baseline demographic and clinical characteristics of participants.**

|  | Experienced surgeon group | Inexperienced surgeon group | P |
|---|---|---|---|
| Number of eyes | 106 | 110 | |
| Number of patients, male/female (n, %) | 66 (62.3%)/40 (37.7%) | 57 (51.8%)/53 (48.2%) | |
| Eye, Right/Left, (n, %) | 54 (51.0%)/52 (49.0%) | 62 (56.4%)/48 (43.6%) | |
| Age, mean (SD), years | 58.7 ($\pm$ 13.4) | 63.6 ($\pm$ 9.7) | 0.003 |
| Preoperative BCVA, median (IQR), decimal | 0.200 (0.016–0.667) | 0.160 (0.010–0.660) | 0.497 |
| Pseudophakic ($n$, %) | 34 (32.1%) | 40 (36.4%) | 0.507 |
| Macula on ($n$, %) | 40 (37.7%) | 45 (40.9%) | 0.633 |
| RD extent, median (IQR), hours | 6 (4–7) | 5 (4–7) | 0.156 |
| Patients with inferior retinal break ($n$, %) | 41 (38.7%) | 42 (38.2%) | 0.940 |
| Number of retinal breaks, median (IQR) | 2.00 (1.00–3.00) | 2.00 (1.00–3.25) | 0.889 |
| Follow-up, median (IQR), months | 9.65 (5.43–17.80) | 8.48 (5.33–15.47) | 0.325 |

**Notes.**
SD, standard deviation; BCVA, best-corrected visual acuity; IQR, interquartile range; RD, retinal detachment.

**Table 2  Surgical techniques and tamponade selection.**

|  | Experienced surgeon group (106 eyes) | Inexperienced surgeon group (110 eyes) | P |
|---|---|---|---|
| Use of perfluorodecaline, n, (%) | 6 (5.7%) | 10 (9.1%) | 0.336 |
| Retinotomy, n (%) | 8 (7.6%) | 6 (5.5%) | 0.532 |
| 360° retinopexy (%) | 12 (11.3%) | 20 (18.2%) | 0.156 |
| Tamponade, C3F8 (%)/SF6 (%) | 46 (43.4%)/60 (56.6%) | 67 (60.9%)/43 (39.1%) | <0.001 |
| Sclerotomy suture, n (%) | 1 (0.9%) | 8 (7.3%) | 0.020 |

**Notes.**
C3F8, perfluoropropane; SF6, sulfur hexafluoride.

**Table 3  Results and complication rates.**

|  | Experienced surgeon group (106 eyes) | Inexperienced surgeon group (110 eyes) | P |
|---|---|---|---|
| Single surgery success rate, n (%) | 100 (94.3%) | 103 (93.6%) | 0.828 |
| Final success rate, n (%) | 106 (100%) | 110 (100%) | |
| Postoperative BCVA, median (IQR), decimal | 0.800 (0.500–1.000) | 0.800 (0.500–1.000) | 0.983 |
| BCVA change, mean (SD), decimal | 0.348 ($\pm$ 0.363) | 0.405 ($\pm$ 0.335) | 0.234 |
| Postoperative BCVA, macula on, median (IQR), decimal | 1.000 (0.667–1.000) | 1.000 (0.800–1.000) | 0.868 |
| Postoperative BCVA, macula off, median (IQR), decimal | 0.667 (0.333–1.000) | 0.667 (0.500–1.000) | 0.877 |
| Complication rate –total, n (%) | 69 (65.1%) | 75 (68.2%) | 0.630 |
| Complication rate –cataract, n (%) | 48 (66.7%) | 56 (80.0%) | 0.073 |
| Complication rate –intraocular hypertension, n (%) | 34 (32.1%) | 38 (34.5%) | 0.700 |
| Complication rate –iatrogenic retinal break, n (%) | 4 (3.8%) | 4 (3.6%) | 0.957 |
| Complication rate –other, n (%) | 6 (5.7%) | 9 (8.2%) | 0.466 |

**Notes.**
SD, standard deviation; BCVA, best-corrected visual acuity; IQR, interquartile range.

median final BCVA in patients who required two surgeries was 0.417 (IQR 0.250–0.500) decimal. The median final BCVA in patients who required three surgeries was 0.250 (IQR 0.188–0.250) decimal.

The complication rate was similar between both groups. The most common complication was the progression of cataract in phakic patients who required cataract surgery. All these patients underwent cataract surgery before the end of follow-up. The second most common complication was postoperative intraocular hypertension. Three patients in each group required temporary therapy with oral acetazolamide, and one patient in the ESG underwent laser iridotomy. All other cases were resolved with IOP-lowering topical medication. Other complications included intraocular hypotony in one patient in each group that resolved without therapy. Two patients in the ESG and four patients in the ISG underwent further PPV for ERM formation in the macula. A postoperative macular hole occurred in two patients in the ESG and one patient in the ISG. Subretinal perfluorocarbon occurred in one patient in ESG. In ISG, one patient suffered an iatrogenic posterior lens capsule tear, and one patient underwent further PPV for intraocular haemorrhage.

## DISCUSSION

In our study, inexperienced surgeons were able to match the success rate of an experienced surgeon in uncomplicated RRD from the very start, suggesting a short learning curve. There is a possible bias in our study in the patient selection in each group. It may be argued that more experienced surgeons would operate on more complex cases. It is impossible to eliminate selection bias in retrospective studies. We tried to minimise it by including only patients with no or only mild PVR. We also excluded patients where silicon oil was used as tamponade and patients operated using 23-gauge vitrectomy, because in our clinic, there is a tendency to use both in more complex cases. The only statistically significant difference between the two groups was in the age of the patients. We do not believe it influenced the results since all the other anatomical and functional characteristics of retinal detachment were similar in both groups. It should also be noted that 68% of all the PPV for retinal detachment performed by inexperienced surgeons met all the inclusion and exclusion criteria and were included in the study. Also, scleral buckling is not performed in our clinic, the only alternative to PPV for the treatment of RRD in our clinic is pneumatic retinopexy.

Another factor that might have influenced the results of the study is the difference in manual skills between surgeons. However, this factor is hard to quantify. Differences in manual skills might affect the time of surgery, the incidence of iatrogenic retinal tears, iatrogenic posterior lens capsule tears, cases of lens touch in phakic patients, and single surgery success rates. We evaluated the incidence of iatrogenic retinal tears, iatrogenic posterior lens capsule tears and single surgery success rates and found no difference between individual surgeons.

Single operation success rates in both groups were comparable to previously published figures for 25 g PPV with gas tamponade (*Miller et al., 2008*; *Mura, Tan & De Smet, 2009*; *Kunikata & Nishida, 2010*; *Gotzaridis et al., 2016*). We believe this is due to the simplified

operation technique used in our clinic. We try to limit surgical techniques that have not been shown to improve uncomplicated RRD surgery outcomes. These include the use of perfluorodecalin (*Schneider, Geraets & Johnson, 2012*; *Vidne et al., 2018*), complete subretinal fluid drainage (*Yamaguchi, Ataka & Shiraki, 2014*; *Gotzaridis et al., 2016*; *Chen et al., 2017*), and 360° retinopexy (*Bilgin et al., 2019*). Limiting the use of these techniques may also prevent certain postoperative complications (*Wilbanks et al., 1996*; *Cauchi, Azuara-Blanco & McKenzie, 2005*; *Bouheraoua et al., 2015*; *Liu, Gao & Liang, 2018*).

It should be noted that there is no widely accepted definition of an experienced and an inexperienced surgeon. Different definitions were used in studies comparing the results of experienced and inexperienced surgeons. In the study by *Dugas et al. (2009)*, less experienced surgeons were defined as having performed less than 50 PPV, and more experienced as having performed more than 3,000 PPV. In the study by *Heimann et al. (2006)*, three groups of surgeons were defined. Surgeons with more than 100 PPV were labelled as "specialists", with 31 to 100 PPV as "non-beginners", and those with 30 or fewer PPV as "beginners". In the study by *Ehrlich et al. (2011)*, beginner surgeons were vitreoretinal fellows, in the study by *Sallam et al. (2018)*, the results of consultant surgeons, independent non-consultant surgeons and trainees (fellows and specialist registrars) were compared. In the study by *Keller, Haynes & Sparrow (2016)*, trainees with less than 12 months' experience were compared with senior surgeons who had completed 30 or more operations in the audit period. In our study, both beginner surgeons performed less than 30 PPV before the start of the inclusion period. Both were trained in PPV two years prior; however, it should be noted that in the Czech Republic, there are no institutionalised subspecialty education programs that would be comparable to fellowship programs.

The final success rate was 100% in both groups. This might have been thanks to the low number of reoperations—only 13 patients had retinal redetachment after the first surgery. Also, the time to surgery from initial diagnosis of retinal detachment was very short in both groups (median 2.0 days in ESG, 1.0 day in ISG). Another factor might have been the short duration of retinal detachment in both groups, with a median duration of symptoms of only 5.0 days. Also, all patients were instructed on the normal postoperative course of events and were encouraged to come for a check-up immediately in case of abnormal development. The first check-up was usually scheduled one month after the surgery. We believe these factors led to early diagnosis of retinal redetachment and prompt reoperation, which lowered the risk of complications that might have led to unfavourable outcomes, like the development of PVR.

Surgical techniques employed by the experienced and inexperienced surgeons in our study were similar, with the exception of the selection of intraocular tamponade and the frequency of sclerotomy suturing. Inexperienced surgeons used C3F8 gas more frequently, a longer-lasting and more "secure" option. We believe this was due to inexperience and subsequent lack of self-confidence, which led inexperienced surgeons to use C3F8 gas even when it might not have been necessary. In our opinion, the more frequent use of C3F8 in ISG was also the reason for the higher cataract surgery rate after the PPV between the ESG and ISG, although the difference was not statistically significant. The higher rate of sclerotomy suturing could possibly be explained by the longer operating times in the ISG;

however, we do not have data to support this claim, as the duration of the operation was not recorded.

The complication rate was similar in both study groups. There was a notable difference in cataract surgery rate after the PPV between the ESG and ISG, although the difference was not statistically significant. Apart from the more frequent use of C3F8 gas for tamponade, the patient age difference between the ISG and the ESG might have been a contributing factor. The low postoperative intraocular hypotony rate can be explained by the oblique cannula insertion technique used by all surgeons (*Acar et al., 2008*; *Bourgault & Tourville, 2012*) and the use of digital palpation to assess IOP after the removal of cannulas. However, it should be noted that some complications were not evaluated. These include postoperative cystoid macular oedema, as optical coherence tomography (OCT) was not performed regularly in all patients during the follow-up visits.

Other studies have shown comparable results in PPV for RRD between experienced and inexperienced surgeons (*Heimann et al., 2006*; *Dugas et al., 2009*; *Ehrlich et al., 2011*; *Mazinani et al., 2012*; *Keller, Haynes & Sparrow, 2016*; *Sallam et al., 2018*). *Ehrlich et al. (2011)* also compared the surgical techniques employed by vitreoretinal fellows and consultants. Compared to our study, they used perfluorocarbon more frequently—in 28.3% of eyes in the vitreoretinal fellow group and 32% of eyes in the consultant group, compared to 9.1% in the ISG and 5.7% in the ESG in our study. Interestingly, the vitreoretinal fellows in their study tended to use SF6 gas more frequently than the consultants, as opposed to our study. The study does not report the extent of retinopexy, the number of eyes requiring sclerotomy sutures, nor the number of retinotomies.

*Dugas et al. (2009)* and *Ehrlich et al. (2011)* compared the success rates of sutureless PPV for RRD between experienced and inexperienced surgeons and used similar exclusion and inclusion criteria as our study. In both these studies, the combined single operation success rate was 75% (75.4% and 80.9%, respectively, for experienced surgeons and 74.8% and 70.0%, respectively, for inexperienced surgeons). This was significantly lower than in our study. This can be explained by the improvements in surgical instruments, vitrectomy machines, and surgical techniques. For example, high-speed vitrectomy has been shown to lower the number of iatrogenic retinal tears during PPV (*Rizzo, Genovesi-Ebert & Belting, 2011*). Other reasons could be a low incidence of different factors that make RRD surgery more challenging, such as RRD in highly myopic eyes, giant retinal tears, *etc.*, in our study. On the other hand, it is worth noting that in a study by *Ehrlich et al. (2011)*, less experienced vitrectomy surgeons were recruited from fellows who had extensive experience in other intraocular procedures. In contrast, in our study, both inexperienced surgeons had no previous experience with intraocular surgery.

As for the impact of our study on clinical practice and surgical training curricula, we believe that with a supportive teaching environment, close observation by the teaching surgeon at the beginning of the training period and constructive feedback, the learning curve for PPV can be quite short. In our study, both inexperienced surgeons without any previous experience with intraocular surgery spent 2 years in training before performing their first complete vitrectomy. However, this was relatively low-intensity training with only one day a week spent at the operating theatre. We believe that with higher-intensity training,

 

this period can be significantly reduced. The training time can be reduced further with modern technological means, both in the operating theatre and outside. In the operating theatre, 3D visualisation systems can significantly speed up and simplify teacher-trainee interaction and communication (*Razavi et al., 2023*). Outside the operating theatre, virtual reality training and modern simulators can prepare trainees for different scenarios they can encounter during surgery (*Deuchler et al., 2016*).

Both inexperienced surgeons matched the results of an experienced vitreoretinal surgeon in PPV for uncomplicated RRD. This fact can benefit the patients, because uncomplicated RRD usually benefit most from prompt intervention and surgeon unavailability is one of the factors that can lead to delays.

The limitations of our study include a low number of evaluated surgeons, the retrospective nature of our study and a subsequent lack of randomisation. Besides the selection bias, other problems associated with retrospective studies could have influenced our results. These include incomplete reporting on some observed variables, namely, time from diagnosis to surgery and duration of symptoms. Furthermore, the minimal follow-up interval in our study was 3 months, which might have caused the underestimation of the number of subsequent retinal detachments. However, according to previously published reports, most retinal redetachments occur within 3 months after the initial surgery (*Lee et al., 2014*; *Heath Jeffery et al., 2020*).

## CONCLUSIONS

A 25 g PPV with gas tamponade for the treatment of RRD yields excellent anatomical results and improvement in BCVA. With good technique and the use of modern vitrectomy machines and instruments, some inexperienced surgeons can achieve a high single-surgery success rate, suggesting a short learning curve. The complication rate is comparable between experienced and inexperienced surgeons.

## ACKNOWLEDGEMENTS

Grammarly AI-powered software (Grammarly Inc., USA) was used for language corrections and editing of the manuscript.

### Funding

The authors received no funding for this work.

### Competing Interests

The authors declare there are no competing interests.

### Author Contributions

- Martin Pencak conceived and designed the experiments, performed the experiments, analyzed the data, prepared figures and/or tables, authored or reviewed drafts of the article, and approved the final draft.

- Zbynek Stranak performed the experiments, authored or reviewed drafts of the article, and approved the final draft.
- Jakub Dite analyzed the data, prepared figures and/or tables, and approved the final draft.
- Jana Vranova analyzed the data, prepared figures and/or tables, and approved the final draft.
- Pavel Studeny conceived and designed the experiments, authored or reviewed drafts of the article, and approved the final draft.
- Miroslav Veith performed the experiments, authored or reviewed drafts of the article, and approved the final draft.

### Human Ethics

The following information was supplied relating to ethical approvals (i.e., approving body and any reference numbers):

Approval was granted by the Ethics Committee of Third Faculty of Medicine, Charles University, Prague and the Kralovske Vinohrady University Hospital, Prague under the number EK-R/01/0/2020.

### Data Availability

The raw data are available in the Supplementary File.

### Supplemental Information

Supplemental information for this article can be found online at http://dx.doi.org/10.7717/peerj.19795#supplemental-information.

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
