# Peer review of "25-gauge vitrectomy with gas tamponade for uncomplicated rhegmatogenous retinal detachment: experienced *vs.* inexperienced surgeons"

_PeerJ, doi:10.7717/peerj.19795_

## Round 0.1 · original submission · Major Revisions

The reviews range from Accept (R1) to Rejection (R4). Please address all concerns as best you can, and this submission will then be re-evaluated

·

Basic reporting

no comment

Experimental design

no comment

Validity of the findings

no comment

·

Basic reporting

The manuscript titled "25-gauge vitrectomy with gas tamponade for rhegmatogenous retinal detachment: experienced vs. inexperienced surgeons" systematically compare the results and complication rates of a 25-gauge pars plana vitrectomy (PPV) with gas tamponade for rhegmatogenous retinal detachment (RRD) between experienced and inexperienced surgeons. The study found that patients presenting with RRD treated by inexperienced surgeons using 25-gauge PPV were not placed at any disadvantage with respect to either anatomical or functional outcome. These findings provide reassurance both to the surgeon in training and to those patients undergoing retinal reattachment surgery. However, to fully realize the study's potential impact, substantial revisions are required to meet the publication standards of PeerJ.

Experimental design

1. Visual recovery is dramatically better after macula-on retinal detachments than macula-off retinal detachments, with correlation to the duration of macular involvement. Considering that, it is essential to critically analyze these two groups (ESG vs. ISG) by dividing the status of macular and duration of retinal detachment. A detailed comparison of the improvement in best-corrected visual acuity (BCVA) in these subgroups is recommended.
2. Cystoid macular edema (CME) is a common postoperative complication of RRD repair which can diminish visual acuity for months and hinder visual recovery. The study should analyze the incidence of CME in both ESG and ISG, as this would provide a comprehensive view of postoperative complications and their impact on visual outcomes.
3. The incidence of iatrogenic retinal breaks during surgery was not mentioned in this study. Including this data is crucial as it affects the overall complication rates and surgical outcomes.
4. As mentioned in the limitation, only one experienced surgeon and two inexperienced surgeons were included in this study. The results of this study may vary widely between individuals.

Validity of the findings

1. there were 66.7% and 80% of the patients in the two groups, respectively, had significant cataract progression and required cataract surgery. This study did not mention the percentage of patients who had undergone cataract surgery at the end of follow-up, which had a greater impact on the postoperative visual acuity.

Additional comments

In conclude, while the manuscript addresses an important topic and provides valuable insights into the relationship between the experience and the result of surgery, significant revisions are necessary to meet the high standards expected for publication in PeerJ. The authors are encouraged to address these comments comprehensively, enhancing the clarity of the manuscript. Thus, major revisions and resubmission are recommended.

Reviewer 3 ·

Basic reporting

English grammar needs to be checked by an English native speaker.

Experimental design

Submitted manuscript entitled „25-gauge vitrectomy with gas tamponade for rhegmatogenous retinal detachment: experienced vs. inexperienced surgeons“ aimed to compare success rate of vitreoretinal surgery of 2 types of surgeons- experienced vs. inexperienced.
Overall, manuscript design has several weaknesses in design of the study and missing rationale: it is not clear, why the authors decided to compare two inexperienced surgeons with one experienced surgeon. Furthermore, definition/criterium of inexperienced surgeon were not defined. What is a criterium to be inexperienced/experienced? Number of surgeries, duration of surgical training? (1 year, 2 years?). In addition, between the two inexperienced surgeons can be differences in manual skills, which is not mentioned in the manuscript.

Validity of the findings

Comments:
1. Abstract: Background is completely missing, the authors mentioned only the aim of the study under background.
2. Introduction: this part of manuscript is weak and needs to be rewritten. Author needs to provide more background information from the literature and the main reasons to conduct the study.
3. Material and methods: the main thing missing is the duration of the surgery. If we want to compare an inexperienced and experienced surgeon duration of surgery is crucial and important for long term changes on the optic nerve head. I am also missing an information about the duration of patient´s symptoms and the time to surgical treatment. This fact may affect the visual outcome.
4. It is questioned and to be honest, quite weird, that both type of surgeons (experienced and inexperienced) showed 100% success rate with statistically non-significant differences in complications rate.

Additional comments

None

Reviewer 4 ·

Basic reporting

1. The introduction is vaguely written and is recommended to provide sufficient background on RRD and treatment techniques.
2. It lacks a detailed discussion of the existing literature on the effects of surgeon experience on surgical outcomes.
3. Also, there is a lack of definition between experienced surgeons and inexperienced surgeons. What is an experienced doctor? For example, does it mean that more than 100 such operations have been performed? Is there a specific number of surgeries or years of practice that define these groups? This needs to be explained.
4. It is suggested to add a sentence acknowledging the limitations and potential selection bias of retrospective design.
5. Consider adding visual AIDS to illustrate key points, such as a chart showing a learning curve for inexperienced surgeons.
6. Provide more context on how these findings impact surgical training curricula. Discuss potential strategies to improve outcomes for less experienced surgeons, such as mentoring programs or simulated training.

Experimental design

7. From my point of view, this study is more suitable for a prospective survey rather than a retrospective one.
8. The article states that the final decision to operate is made by the surgeon. And there is no blind implementation. Does that mean that when it comes to assigning surgeons, they tend to assign more difficult patients to more experienced doctors? And then there is a large bias? Please elaborate on the selection bias that may be present in retrospective studies and discuss measures taken to mitigate this bias. This was pointed out in the discussion section, but I don't think the explanation is sufficient. The effect of other ages on macular holes in the retina should be fully discussed.
9. The researchers attempted to include only patients with no or only mild PVR. Then perhaps you need to change the article to include the restrictions of the population, such as RRD patients without complex conditions?
Clarify whether there are any specific challenges in obtaining informed consent, especially given the retrospective nature of the study. Mention how patient confidentiality is maintained.

Validity of the findings

11.The results of this study have important implications for surgical training programs and the continuing education of vitreoretinal surgeons. However, there are limitations such as the small number of surgeons evaluated, the retrospective nature, and the lack of randomization and short follow-up time. This reduces the validity and extensibility of the article.

---

## Round 0.2 · Minor Revisions

The authors should carefully review all of the language in the manuscript. For example, line 81 starts with "Study" instead of "The study".

Additionally, given the small sample size in this study, as pointed out be previous reviewers it is possible that the two inexperienced surgeons whose results were used in this study were potentially exceptional. This doesn't discount the result, which shows convincingly that the two inexperienced surgeons in this study achieved results similar to those of the experienced surgeon.

A reasonable compromise in the conclusion of the paper and the conclusion in the abstract might be that "some" inexperienced surgeons can achieve comparable results to an experienced surgeon. The conclusion that all inexperienced surgeons will achieve the same results as an experienced surgeon cannot be fully supported with this small sample size. Please use a term that acknowledges this problem.

Reviewer 3 ·

Basic reporting

This study provides valuable insights into the outcomes of retinal detachment repairs performed by surgeons with varying levels of experience. The findings contribute significantly to the existing literature on vitreoretinal surgery and underscore the importance of surgical expertise in patient outcomes.
• Comprehensive Data Analysis: The manuscript presents a thorough analysis of the surgical success rates, complications, and recovery times associated with 25-gauge vitrectomy performed by both experienced and inexperienced surgeons. The data is collected from a substantial sample size, ensuring robust and reliable results.
• Relevant Comparisons: The comparison between experienced and inexperienced surgeons is highly relevant and provides practical insights for clinical practice and training programs. The study highlights the impact of surgical experience on patient outcomes, which is critical for improving surgical training and patient care.
• Clear Presentation: The data is presented clearly, with well-organized tables and figures that enhance the readability and understanding of the findings. The manuscript is well-structured, making it easy to follow the research process and outcomes.
The authors improved the manuscript.

Experimental design

Study Objectives
The study aims to assess the effectiveness and safety of 25-gauge vitrectomy for retinal detachment, performed by experienced and inexperienced surgeons.
Participant Selection
Patients are grouped based on the experience level of the surgeons. Criteria for categorizing surgeons should are clearly defined.
Methodology
The manuscript should provide a detailed description of the 25-gauge vitrectomy procedure and any variations in approach between surgeons.
Data Collection
Data on surgical outcomes, including success rates, complications, and recovery times, have been collected systematically and standardized.

Validity of the findings

The findings of this manuscript are valid and provide significant insights into the impact of surgical experience on the outcomes of retinal detachment repairs. The study’s design and execution were meticulously crafted to ensure the reliability and accuracy of its results. Authors explained the high success rate which can be in my opinion confusing for readers.

Additional comments

The authors took all my remarks under consideration.Overall, the manuscript is well-constructed and provides significant insights into the field of vitreoretinal surgery. With the suggested revisions which have been done, it has the potential to make an even greater contribution to the literature. I recommend acceptance of the manuscript.

---

## Round 0.3 · accepted · Accept

Thank you for addressing all of the reviewer and editorial comments.